**Brief Communication**

# Evidence of monkeypox virus clade IIb lineage A.2.2 in the Republic of the Congo and co-circulation of clade Ia, Ib and clade IIb

Felix Koukouikila-Koussounda[1,2,7], Claude Kwe Yinda [3,7], Pembe Issamou Mayangue[1,2], Dachel Aymard Eyenet Boussam[1], Reiche Golmard Elenga[1,2], Leadisaelle Hosanna Lenguiya[1,2], Ghislain Dzeret Indolo[1], Bani Reize Vishnou Ampiri[1], Lucette Nathalie Macosso[1], Igor Judicaël Louzolo[1], Isaac Samuel Onyakouang[1], Jordy Exaucé Lyelet Demboux [1], Jean Medar Kankou[4], Aristide Gilbert Nianga[5], Avelin F. Aghokeng [6], Vincent Jacobus Munster [4,8] & Fabien Roch Niama [1,2,8]

In 2022, monkeypox virus (MPXV) clade IIb emerged resulting in a global epidemic driven by human-to-human transmission, mostly through sexual contact mainly among the population of men who have sex with men. To date, published data on the circulation of MPXV clade IIb in the central African region are absent. Here we describe a case of laboratory-confirmed mpox clade IIb lineage A2.2 in Pointe-Noire, the second largest city of the Republic of the Congo. Whole-genome phylogenetic analysis placed the MPXV in clade IIb, lineage A.2.2 currently emerging in West Africa, in particular Sierra Leone. An additional 16 cases of clade Ia, 32 cases of clade Ib and one additional introduction of clade IIb were identified by passive surveillance in the Republic of the Congo in 2025. The detection of clade IIb mpox marks the third distinct MPXV clade and lineage co-circulating in the human population, together with clade Ia and Ib. This underscores the need for improved surveillance and diagnostic strategies to identify the respective clade and lineage circulating in the human population. Strengthening of regional capacity for case detection, contact-tracing, public health measures and affordable vaccines are urgently needed to reduce the global risk for both clade I and clade II MPXV.

Mpox is a disease endemic in central and west African regions[1]. The disease is caused by two different clades of monkeypox virus (MPXV), I and II, which are further subdivided into subclades Ia and Ib, and IIa and IIb[2,3]. Historically, clade I MPXV circulation was observed in Democratic Republic of the Congo (DRC), the Republic of the Congo (RoC), Central African Republic, South Sudan, Gabon and Cameroon, whereas clade II circulation was observed in Sierra Leone, Liberia, Ivory Coast, Ghana, Benin, Cameroon and Nigeria[4,5]. Cameroon has historically been the only country in which circulation of both clade I and clade II MPXV has been observed[6].

Clade IIb emerged in early May 2022 causing a global epidemic of mpox[1]. The clade IIb outbreak was characterized by rapid expansion of the disease in nonendemic countries, especially in Europe and North America, driven by sexual transmission among men who have sex with men[4].

In August 2024, because of the rapidly rising number of mpox cases caused by clade Ia and Ib in the DRC and neighboring countries[3,7],

**Fig. 1 | Map of the RoC showing departments with MPXV-positive cases in 2025.** Slice size in the pie charts represents the number of mpox cases, and the colors indicate the number of cases per subclade. There were 37 cases in Brazzaville (4 clade Ia, 30 clade Ib and 3 unknowns); 5 cases in Cuvette-Ouest all clade Ia; 8 cases in Pointe-Noire (1 clade Ia, 2 clade Ib, 2 clade IIb and 3 unknowns); and 6 cases in Cuvette, all clade Ia. Basemap data from Natural Earth (https://www.naturalearthdata.com/).

Africa CDC listed mpox as a Public Health Emergency of Continental Security. Subsequently, the World Health Organization declared mpox a Public Health Emergency of International Concern for the second time since 2022[8,9].

Currently West Africa, in particular Sierra Leone, is facing a major mpox clade IIb outbreak of a new clade IIb lineage A.2.2.1 virus derived from Nigerian A.2.2 viruses[10,11]. Similar to what has been observed for clade Ib, transmission by heterosexual contact appears to be one of the drivers of this epidemic[12–15].

RoC has experienced a dramatic increase in mpox cases over the past two years[16]. Historically, only clade Ia has caused outbreaks in RoC, primarily in the northern part of the country. However, several introductions of clade Ib in the RoC capital Brazzaville were identified 2024 and 2025, with a clear epidemiological and phylodynamic link to Kinshasa, DRC[17]. Clade Ia and Ib now display characteristics of sustained human-to-human transmission based on their apobec3 signatures in the genome. Here we report the introduction of clade IIb mpox in RoC, highlighting the continuous evolving epidemiology of MPXV.

The case was a 43-year-old man from Pointe-Noire, which is the second largest city and the economic capital of RoC. The patient reported a recent travel history to France and Ivory Coast and was identified when visiting a private outpatient clinic on the morning of 22 March 2025 with complaints of fever, asthenia and a skin rash in the genital area. The physician initially suspected syphilis. After close physical examination of the patient, he noticed the presence of disseminated skin lesions on different parts of the body with vesicles (fewer than 15) and pustules on the limbs, trunk, back and buttocks. Diagnostic tests for syphilis, HIV, hepatitis B and hepatitis C were negative. Based on the clinical description, the case was referred as suspected mpox as part of the passive surveillance of the RoC Directorate of Epidemiology

and Disease Control of the Ministry of Health and Population under the framework of the Center des Opérations des Urgences de Santé Publique. Skin lesion and blood samples were collected and transferred to the National Public Health Laboratory in Brazzaville for molecular diagnostics. The patient was isolated at home, given paracetamol and primalan for 5 days, and cyteal for treatment of the skin infection. He made a full recovery after 19 days. No transmission of the MPXV to family members or healthcare workers was observed. The home isolation set-up primarily comprised of voluntary isolation in housing separate from family members, with patient follow-up every 3 days until lesion resolution. Healthcare workers who examined the patient and his direct contacts were told to contact the mpox disease control team in case of any observation of symptoms including myalgia, fever and skin lesions. No mpox symptoms were reported for any of the healthcare workers or direct contacts.

The initial diagnostic result for the suspected mpox case on GeneXpert was MPXV clade II positive. A cycle threshold (Ct) value of 22 was obtained for the skin lesion sample and 34.8 for the blood sample, and nonvariola positive (generic assay to detect all nonvariola orthopox viruses including MPXV clades I and II) with a Ct value of 21.7 for the skin lesion sample and 32.5 for the blood sample. In the second analysis using the RADI Fast MPXV detection kit, the result was positive for clade II with a Ct value of 17 for the skin lesion sample and 33 for the blood sample.

In addition to the above-described case, an additional 56 mpox cases were identified from January 2025 to August 2025 in RoC. Samples were obtained from 7 of the 12 administrative departments of RoC namely, Brazzaville, Pointe-Noire, Likouala, Cuvette, Kouilou, Plateaux and Niari. During this time 16 cases of clade Ia, 32 cases of Ib and two introductions of clade IIb were identified. Epidemiological data suggest that during this period in 2025 several lineages of clade Ia, as well

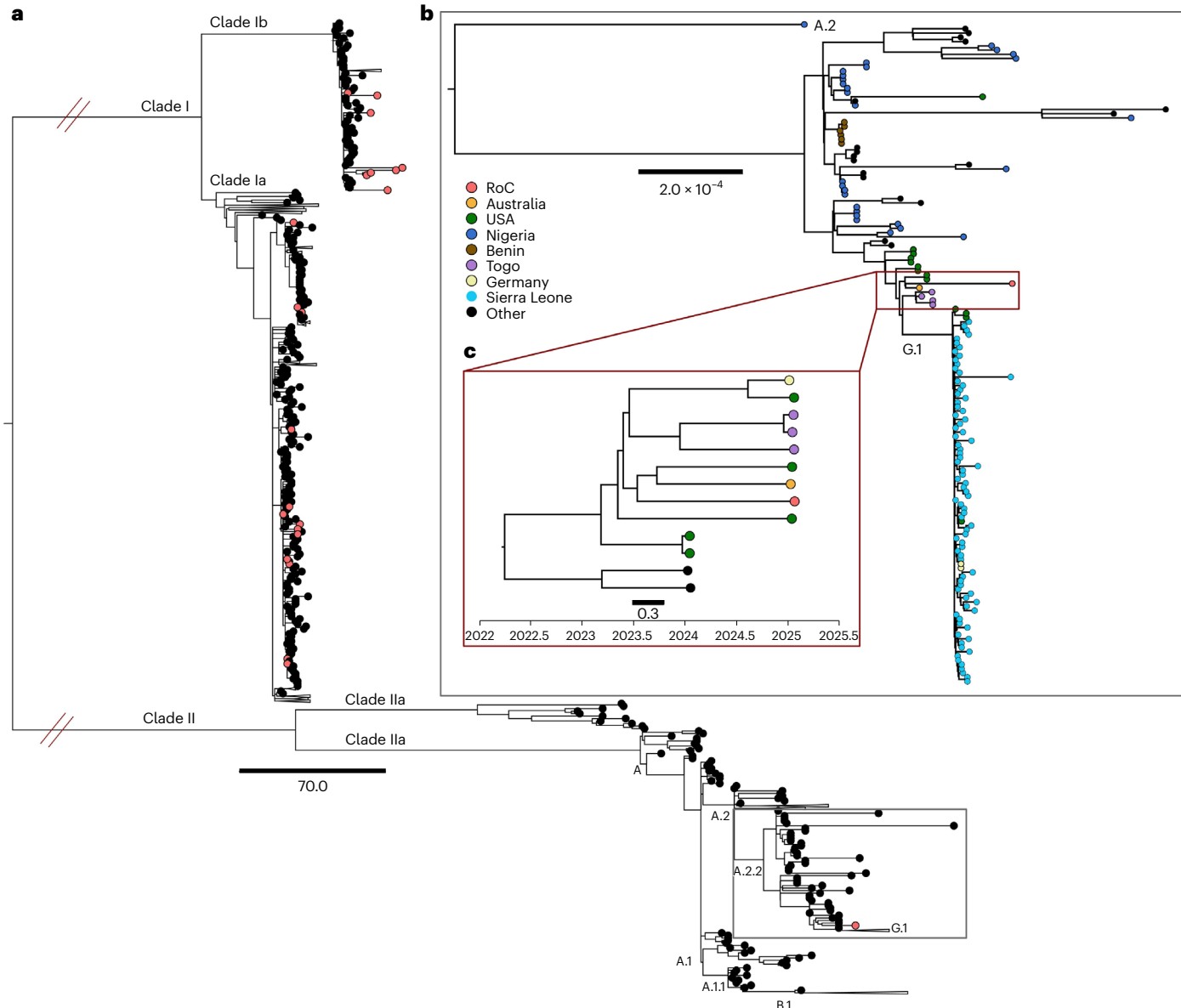

**Fig. 2 | Phylogenetic analysis of MPXV nucleotide sequences of all clades and lineages. a**, Phylogeny including 53 contemporary MPXV genomes of clade Ia and Ib, and clade IIb obtained from surveillance in RoC depicted as red circles in the phylogenetic tree. A complete list of MPXV genomes obtained from the surveillance in RoC is listed in Supplementary Table 2. **b**, Close-up tree of clade IIb, lineage A.2. The colors of the circles represent the origin of the MPXV genome, with clade IIb obtained from surveillance in RoC depicted as red circle in the phylogenetic tree. The tree was visualized in FigTree (http://tree.bio.ed.ac.uk/software/figtree/). Scale bar: nucleotide substitutions per site. **c**. Maximum clade credibility phylogeny of MPXV of a subcluster of clade IIb A.2.2 with time shown on the *x* axis. The Bayesian tree was constructed using a strict clock and exponential model.

as Ib were co-circulating in the major administrative (Brazzaville) and economic (Pointe-Noire) hubs in RoC (Fig. 1).

Whole-genome sequences were obtained from the MPXV-positive specimen, with a genome length of 197,200 base pairs. Phylogenetic analysis identified this sequence as MPXV clade IIb, lineage A.2.2 (Fig. 2a). The Congolese sequence clustered with recent lineage A.2.2 sequences from Togo, the USA and Australia, forming a clade that branches into the new G.1 cluster, which is composed mostly of sequences from Sierra Leone (Fig. 2b). We next used a Bayesian Markov chain Monte Carlo (MCMC) approach to more rigorously assess the evolutionary dynamics of these sequences. Based on the best-fit model (Supplementary Table 1), the sequences showed an estimate for the mean rate of evolution of $5.12 \times 10^{-5}$ (95% highest posterior density (HPD), $6.47 \times 10^{-6}$ to $8.41 \times 10^{-5}$) substitutions per site per year, comparable with the rate of evolution of clade IIb lineage A.2 ($5.53 \times 10^{-5}$

(95% HPD $3.39 \times 10^{-5}$ to $7.46 \times 10^{-5}$)[18]. The time to the most recent common ancestor between the RoC sequence and its closest non-RoC relatives was estimated at mid-2023 (mean 2023.5), with a 95% HPD interval of 2021.44 to 2024.63 (Fig. 2c). These estimates indicate that the lineage leading to the RoC strain originated between approximately August 2024 and March 2025, consistent with a recent cross-border importation event rather than sustained local circulation.

Since 2022, molecular diagnostics for MPXV has been established in RoC allowing for the implementation of a passive mpox surveillance system in the country. In 2023, next-generation sequencing capacity was added, and the passive surveillance system allows for molecular diagnostics of mpox suspect cases, clade and lineage-defining quantitative real-time polymerase chain reaction (qRT-PCR) assays and the ability to perform molecular epidemiological studies using genomic MPXV data.

Our mpox surveillance successfully identified the co-circulation of multiple clade Ia lineages in RoC in 2025, the introduction of clade Ib from the DRC on two independent occasions in 2024 and now the introduction of clade IIb. Data generated on the detection of A.2.2 MPXV clade IIb were directly shared with the RoC Directorate of Epidemiology and Disease Control of the Ministry of Health and Population under the framework of the Centre des Opérations des Urgences de Santé Publique. This resulted in the implementation of mpox control measures in Pointe-Noire including strengthening of local laboratory diagnostics using GeneXpert platforms to increase decentralized MPXV diagnostic testing; enhanced molecular epidemiology surveillance by next-generation sequencing; outreach to hospitals, healthcare settings and healthcare workers to increase awareness on mpox and to improve mpox clinical management; and hospital and healthcare hygiene practices with a focus on nosocomial transmission prevention.

Phylogenetic analysis placed clade IIb MPXV into lineage A.2.2. MPXV clade IIb has been circulating cryptically for nearly a decade in Nigeria. The ongoing circulation of clade IIb in Nigeria, including distinct sub-lineages such as A.2.2, have resulted in recent regional outbreaks in Sierra Leone, as well as travel-related cases outside Africa, including in Germany, the USA and Australia. The rapid spread in West Africa, introduction into Central Africa and travel-related cases outside Africa resemble the rapid spread of the 2022 global mpox outbreak because the A.2 lineage also originates from Nigeria, has spread throughout West Africa and has been detected in Australia, Europe and the USA. Because only a single A.2.2 MPXV clade IIb genome has been detected and sequenced from RoC to date, our inference of MPXV introduction must be interpreted with caution. Sparse sampling and limited genomic surveillance in RoC hinder the reconstruction of transmission chains and phylogeographic origin. Although the RoC strain (hMpxV/Congo/BZV-LNSP-CG051/2025) clusters in clade IIb lineage A.2.2 alongside genomes from the USA and Australia, this placement alone cannot reliably indicate the direction or source of the introduction. The wide 95% HPD interval for the ancestral node (2021.44 to 2024.63) further underscores the uncertainty surrounding the timing and geographic pathway of importation. Expanded sequencing and continuous surveillance are therefore essential to clarify the evolutionary trajectory and transmission dynamics of MPXV in Central Africa.

Nevertheless, the clustering of our sequence with MPXV sequences from West Africa, the USA, Germany and Australia all collected in 2025, and given the known travel history of this case to France and Ivory Coast, suggests that the patient infected with clade IIb MPXV likely contracted the infection while traveling in West Africa or Europe. The rapid rise of clade IIb lineage A.2.2 highlights the vulnerability of the region for undetected circulation of MPXV. It also underscores major gaps in regional preparedness, in particular the availability of diagnostics and the ability to conduct molecular epidemiology.

This study has some limitations. Our patient could not provide comprehensive contact-tracing information to health workers, showing the complexity of tracking imported cases, and this does not allow us to completely reconstruct the exposure and transmission history. In addition, very limited genomic information is available from West Africa, or the new A.2.2 lineage altogether, further hampering efforts to determine the origin of exposure as well as the degree of ongoing circulation in sub-Saharan Africa.

This work presents epidemiological and phylogenetic characteristics of a laboratory-confirmed mpox case with MPXV clade IIb, lineage A.2.2 in RoC. The detection of clade IIb in this region highlights the growing complexity of MPXV epidemiology in a historically endemic region. In 2025, cases of clade Ia, Ib and now clade IIb have been detected in RoC. Current genetic and epidemiological data from RoC suggest that each of these MPXV clades and sub-lineages harbors distinct human-to-human characteristics based on APOBEC3 signatures and epidemiological characteristics[16,17,19].

MPXV evolution is driven by a variety of mechanisms and underlies the relatively rapid genotypic and phenotypic change observed during consecutive mpox outbreaks of the past decade. These mechanisms include acquiring mutations (for example, the APOBEC3 mutations), gene duplication and recombination. The co-circulation of clade Ia, Ib and clade IIb in the human population of a relatively confined geographic area dramatically increases the risk for recombination events between viruses of these different clades, potentially resulting new phenotypes[20].

The co-circulation of clade Ia, Ib and clade IIb in the human population highlights the increasing complexity of MPXV epidemiology and the need for a regionally coordinated response[21]. This response should be targeted at region-specific needs, including expanding access to decentralized diagnostics, enhancing the (molecular) epidemiological capacity and a dramatic increase in vaccine access. Implementation of isolation and contact-tracing strategies, as well as community outreach should be a key component of targeted mpox control[9]. Lastly, our ability to detect emerging and re-emerging pathogens is directly related to the accessibility to healthcare and diagnostic infrastructure. Large parts of RoC are relatively remote, and emphasis on increased surveillance in border regions between RoC, DRC and Central African Republic is needed.

## Online content

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

[1]National Public Health Laboratory, Brazzaville, Republic of the Congo. [2]Faculty of Sciences and Techniques, Marien NGOUABI University, Brazzaville, Republic of the Congo. [3]Laboratory of Virology, Division of Intramural Research, National Institute of Allergy and Infectious Diseases, Hamilton, MT, USA. [4]Center for Public Health Emergency Operations, Ministry of Health and Population, Brazzaville, Republic of the Congo. [5]Directorate of Epidemiology and Disease Control, Ministry of Health and Population, Brazzaville, Republic of the Congo. [6]Research Institute for Development (RID), MIVEGEC, Montpellier, France. [7]These authors contributed equally: Felix Koukouikila-Koussounda, Claude Kwe Yinda. [8]These authors jointly supervised this work: Vincent Jacobus Munster, Fabien Roch Niama. ✉e-mail: vincent.munster@nih.gov; fabien.niama@gmail.com

## Methods

### Inclusion and ethics statement

This project was designed through a longstanding partnership since 2010 between the National Public Health Laboratory, Brazzaville, RoC and the Virus Ecology Section at the Division of Intramural Research, National Institute of Allergy and Infectious Diseases, with the primary goal of developing a genomic pathogen surveillance capacity and the implementation of a scalable high-throughput sequencing pipeline for rapid identification and tracking of emerging infectious diseases. The partnership includes support in the development and implementation of diagnostics for high-consequence viruses, appropriate training in biosafety and specimen handling, and the use of validated inactivation procedures.

The Congolese team members led the epidemiological framework for MPXV detection in RoC, the initial molecular diagnosis and the full genome sequencing. The USA team members provided on-site and remote technological and analyses support. All team members collaborated on data ownership, intellectual property and authorship of publications related to the work. Previous work from this region (both from our team and other researchers, in particular from the DRC) was used to guide the design of this study as well as connect our findings to similar research and has been taken into account in the citations for this manuscript.

### Case identification and sample collection

The case is a resident in Pointe-Noire. Pointe-Noire is divided into six health districts namely Lumumba, Mvou-Mvou, Tie-Tie, Loandjili, Ngoyo and Mongo-Mpoukou. Each health district includes several primary public health centers, one referral public hospital and private healthcare facilities. The patient was identified when visiting a private outpatient clinic in the sanitary district of Lumumba. He presented to the attending physician at the clinic on the morning of 22 March 2025 with complaints of fever, asthenia and a skin rash in the genital area. Initial diagnostic tests for syphilis, HIV, hepatitis B and hepatitis C were negative.

Skin lesion and blood samples were collected and transferred to the National Public Health Laboratory in Brazzaville for molecular diagnostics.

### MPXV molecular detection

Skin lesion and blood samples were inactivated in a class III biosafety cabinet and DNA was extracted using the High Pure PCR Template Preparation kit (Roche), following the manufacturer's instructions. Extracted DNA was kept at −20 °C until MPXV DNA detection. Real-time polymerase chain reaction (PCR) assays were performed using the XpertMpox cartridge (Cepheid) and RADI FAST Mpox detection kit (KH Medical) according to the manufacturers' instructions. The RADI FAST Mpox detection kit included specific primers for MPXV clade I and clade II targets. All assays were run on the QuantStudio 5 Real-Time PCR system (Applied Biosystems).

### Whole-genome sequencing and bioinformatics analysis

Nucleic acids extracted from the clinical samples were amplified using a mpox virus-specific primer set covering the MPXV genome.[21] Amplification products were purified, then prepared for Oxford Nanopore Technologies sequencing using the Native Barcoding Kit 96 v.14 (SQK-NBD114.96) according to the manufacturer's instructions. This method allows the addition of unique sequencing adapters and barcodes to each sample, without additional PCR, thus minimizing amplification bias. PCR products were then quantified using the Qubit instrument with the high-sensitivity Qubit 1X dsDNA assay kit (ThermoFisher Scientific). Barcoded libraries were loaded onto an R10.4.1 flow cell and sequenced using a GridION sequencer (Oxford Nanopore).

### Bioinformatics analysis

A structured bioinformatics pipeline was implemented to analyze raw reads from Oxford Nanopore Technologies, following the steps below. Raw reads were subjected to a preliminary quality check using NanoPlot (v.1.44.1), to generate descriptive statistics (length distributions, quality scores) and identify any anomalies in the data. Residual adapter sequences were automatically detected and removed using Porechop (v.0.2.4), optimizing read quality for subsequent steps. Rigorous filtering was applied using Filtlong (v.0.2.1), retaining only reads of a minimum length of 1,500 bases, with an average quality score above 10 and representing the most reliable 80% of the initial set. The filtered reads were aligned against a reference sequence using Minimap2 (v.2.29-r1283) in map-ont mode, specifically optimized for long reads. The resulting alignments were then sorted and indexed with Samtools (v.1.21) for efficient exploitation. The consensus sequence was generated from the aligned reads using Medaka (v.2.0.1), a deep learning-based tool for correcting errors specific to nanopore reads. The generated consensus sequence was submitted to Nextclade (https://clades.nextstrain.org/) for annotation, phylogenetic classification (clade assignment) and mutational event detection (substitutions, deletions, insertions).

### Phylogenetic and phylodynamic analysis

A preliminary overall phylogenetic tree was constructed using Nextclade including all sequences from RoC and the Nextclade dataset 'Mpox virus (All clades)'. Full-length genomes of historical clade IIb A.2.2 MPXV sequences were obtained from GISAID and Pathoplexus (https://doi.org/10.62599/PP_SS_1090.1) and used to construct a more detailed tree. Sequences were aligned using MAFFT (FFT-NS-1 algorithm), with the best model for distance estimates identified using ModelFinder function as the one with the lowest Bayesian information criterion. A maximum likelihood phylogenetic tree was constructed using IQ-TREE2 and branch support was assessed using ultrafast bootstrap approximation (ufBoot, 1,000 replicates). The tree was visualized in FigTree (http://tree.bio.ed.ac.uk/software/figtree/) and rooted using a clade IIb A.2 sequence. Bars indicate nucleotide substitutions per site.

We attempted to compute the time of origin of the RoC clade IIb lineage A.2.2 using the Bayesian MCMC method. Here, we tested three different temporal and demographic scenarios: strict molecular clock with constant population, SkyGrid coalescent model, and exponential growth rate model with path sampling used to rank the final models. For each analysis, two independent chains of 100 million generations (sampled every 10,000 states) were run to ensure convergence and then combined with appropriate burn-in. Statistical uncertainty was reflected in values of the 95% HPD.

### Map generation

Geographic and epidemiological data were processed using R. Mpox cases were aggregated at the administrative department level in RoC and linked to publicly available geographic shapefiles obtained from Natural Earth (https://www.naturalearthdata.com/). Spatial data handling and mapping were performed using the sf and ggplot2 packages available from CRAN, the Comprehensive R Archive Network (https://cran.r-project.org/). Choropleth maps were generated to display the number of positive cases by department, with proportional symbols indicating the distribution of clade Ib cases. A contextual map of Africa highlighting RoC is included to provide geographic reference.

### Statistics and reproducibility

In this study 56 samples were included of active mpox cases in RoC: 37 in Brazzaville (4 clade Ia, 30 clade Ib and 3 unknowns); 5 in Cuvette-Ouest all clade Ia; 8 cases in Pointe-Noire (1 clade Ia, 2 clade Ib, 2 clade IIb and 3 unknowns); and 6 in Cuvette, all clade Ia. Samples were obtained through passive surveillance via notification of local healthcare clinics. Although the current sample size is sufficient to support the conclusion of local transmission of clade Ia, Ib and clade IIb, they likely represent only a small number of the active cases in RoC for a variety of reasons including mildly symptomatic cases, stigma and limited access to healthcare in certain parts of the country.

For the origin of the RoC lineage of clade IIb A.2.2 using the Bayesian MCMC method, the maximum clade credibility tree was estimated from the posterior distribution of trees with node heights scaled to mean values and posterior probabilities showing the statistical support for individual nodes.

## Ethics

Specimen collection and laboratory confirmation were conducted under the passive national surveillance program. Written informed consent was obtained from the patient for publishing without allowing publication of images. Ethical clearance was obtained from the Congolese Foundation for Medical Research Institutional Ethics Committee (Avis no. 053/CEI/FCRM/2024).

## Reporting summary

Further information on research design is available in the Nature Portfolio Reporting Summary linked to this article.

## Data availability

Whole-genome sequences of the MPXV generated in this study are available at GISAID under accession numbers EPI_ISL_19350687, EPI_ISL_19350688, EPI_ISL_19350689, EPI_ISL_20135135, EPI_ISL_20135139, EPI_ISL_20135142, EPI_ISL_20135132, EPI_ISL_20135141, EPI_ISL_20135126, EPI_ISL_20135133, EPI_ISL_20135123, EPI_ISL_20135128, EPI_ISL_20135138, EPI_ISL_20135136, EPI_ISL_20135131, EPI_ISL_20135124, EPI_ISL_20135140, EPI_ISL_20135130, EPI_ISL_20135122, EPI_ISL_20135134, EPI_ISL_20135129, EPI_ISL_20135127, EPI_ISL_20135125, EPI_ISL_20135137, EPI_ISL_19842849.

## Acknowledgements

This work received no external funding and was fully supported by the National Public Health Laboratory. C.K.Y. and V.J.M. were supported by the Intramural Research Program of the National Institutes of Health (NIH). The contributions of the NIH authors were made as part of their official duties as NIH federal employees, are in compliance with agency policy requirements, and are considered Works of the United States Government. However, the findings and conclusions presented in this paper are those of the authors and do not necessarily reflect the views of the NIH or the US Department of Health and Human Services. We thank all the healthcare workers who were involved in identification the case, epidemiological information, collection of specimens and transfer of samples to the NPHL. We express our gratitude to Africa CDC, WHO Congo and CDC Atlanta for continuous support.

## Author contributions

F.K.-K., D.A.E.B., R.G.E. and F.R.N. conceived and designed the study. F.K.-K., D.A.E.B., R.G.E., G.D.I., B.R.V.A., L.H.L. and I.S.O. processed samples and generated the data. C.K.Y., V.J.M., P.I.M., L.N.M., I.J.L., J.E.L.D. and F.R.N. accessed and verified all the data. F.K.-K., D.A.E.B., F.R.N. and C.K.Y. curated, analyzed and interpreted data. F.K.-K., D.A.E.B., C.K.Y., V.J.M. and R.G.E. wrote the first draft of the manuscript. I.S.O., J.M.K., A.G.N., P.I.M., A.F.A., C.K.Y., V.J.M. and F.R.N. revised the manuscript. All authors provided critical revision of the manuscript. All authors had full access to all the data of the study and had final responsibility for the decision to submit for publication.

## Competing interests

The authors declare no competing interests.

## Additional information

**Correspondence and requests for materials** should be addressed to Vincent Jacobus Munster or Fabien Roch Niama.

# Reporting Summary

## Statistics

For all statistical analyses, confirm that the following items are present in the figure legend, table legend, main text, or Methods section.

| n/a | Confirmed | |
|---|---|---|
| ☐ | ☒ | The exact sample size (*n*) for each experimental group/condition, given as a discrete number and unit of measurement |
| ☒ | ☐ | A statement on whether measurements were taken from distinct samples or whether the same sample was measured repeatedly |
| ☒ | ☐ | The statistical test(s) used AND whether they are one- or two-sided *Only common tests should be described solely by name; describe more complex techniques in the Methods section.* |
| ☒ | ☐ | A description of all covariates tested |
| ☒ | ☐ | A description of any assumptions or corrections, such as tests of normality and adjustment for multiple comparisons |
| ☒ | ☐ | A full description of the statistical parameters including central tendency (e.g. means) or other basic estimates (e.g. regression coefficient) AND variation (e.g. standard deviation) or associated estimates of uncertainty (e.g. confidence intervals) |
| ☒ | ☐ | For null hypothesis testing, the test statistic (e.g. *F*, *t*, *r*) with confidence intervals, effect sizes, degrees of freedom and *P* value noted *Give P values as exact values whenever suitable.* |
| ☒ | ☐ | For Bayesian analysis, information on the choice of priors and Markov chain Monte Carlo settings |
| ☒ | ☐ | For hierarchical and complex designs, identification of the appropriate level for tests and full reporting of outcomes |
| ☒ | ☐ | Estimates of effect sizes (e.g. Cohen's *d*, Pearson's *r*), indicating how they were calculated |

*Our web collection on statistics for biologists contains articles on many of the points above.*

## Software and code

Policy information about availability of computer code

| Data collection | NanoPlot (v.1.44.1),Porechop (v.0.2.4), Filtlong (v.0.2.1), Minimap2 (v.2.29-r1283) |
|---|---|
| Data analysis | Medaka (v.2.0.1),BEAST (version 10.05.0) 26 |

For manuscripts utilizing custom algorithms or software that are central to the research but not yet described in published literature, software must be made available to editors and reviewers. We strongly encourage code deposition in a community repository (e.g. GitHub). See the Nature Portfolio guidelines for submitting code & software for further information.

## Data

Policy information about availability of data

All manuscripts must include a data availability statement. This statement should provide the following information, where applicable:
- Accession codes, unique identifiers, or web links for publicly available datasets
- A description of any restrictions on data availability
- For clinical datasets or third party data, please ensure that the statement adheres to our policy

Genbank Accession numbers will be made available upon publication

# Research involving human participants, their data, or biological material

Policy information about studies with human participants or human data. See also policy information about sex, gender (identity/presentation), and sexual orientation and race, ethnicity and racism.

| | |
|---|---|
| Reporting on sex and gender | sex has been used (male) |
| Reporting on race, ethnicity, or other socially relevant groupings | NA |
| Population characteristics | NA |
| Recruitment | Specimen collection and laboratory confirmation were conducted under the passive national surveillance program. Written informed consent was obtained from the case without allowing publication of images. |
| Ethics oversight | Ethical clearance was obtained from the Congolese Foundation for Medical Research Institutional Ethics Committee (Avis n° 053/CEI/FCRM/2024). |

Note that full information on the approval of the study protocol must also be provided in the manuscript.

# Field-specific reporting

Please select the one below that is the best fit for your research. If you are not sure, read the appropriate sections before making your selection.

☒ Life sciences  ☐ Behavioural & social sciences  ☐ Ecological, evolutionary & environmental sciences

For a reference copy of the document with all sections, see nature.com/documents/nr-reporting-summary-flat.pdf

# Life sciences study design

All studies must disclose on these points even when the disclosure is negative.

| | |
|---|---|
| Sample size | 37 |
| Data exclusions | no data excluded |
| Replication | NA |
| Randomization | NA |
| Blinding | Not relevant for clinical specimen |

# Reporting for specific materials, systems and methods

We require information from authors about some types of materials, experimental systems and methods used in many studies. Here, indicate whether each material, system or method listed is relevant to your study. If you are not sure if a list item applies to your research, read the appropriate section before selecting a response.

## Materials & experimental systems

| n/a | Involved in the study |
|---|---|
| ☒ | ☐ Antibodies |
| ☒ | ☐ Eukaryotic cell lines |
| ☒ | ☐ Palaeontology and archaeology |
| ☒ | ☐ Animals and other organisms |
| ☐ | ☒ Clinical data |
| ☒ | ☐ Dual use research of concern |
| ☒ | ☐ Plants |

## Methods

| n/a | Involved in the study |
|---|---|
| ☒ | ☐ ChIP-seq |
| ☒ | ☐ Flow cytometry |
| ☒ | ☐ MRI-based neuroimaging |

## Clinical data

Policy information about clinical studies

All manuscripts should comply with the ICMJE guidelines for publication of clinical research and a completed CONSORT checklist must be included with all submissions.

| | |
|---|---|
| Clinical trial registration | Ethical clearance was obtained from the Congolese Foundation for Medical Research Institutional Ethics Committee (Avis n°053/CEI/FCRM/2024). |
| Study protocol | NA, passive disease surveillance program |
| Data collection | data ollection performed by local epidemiological teams |
| Outcomes | NA |

## Plants

| | |
|---|---|
| Seed stocks | NA |
| Novel plant genotypes | NA |
| Authentication | NA |

