## [Peer Review File · Nature Medicine]

First laboratory-confirmed case of monkeypox virus clade IIb lineage A.2.2 in the Republic of the Congo, co-circulation of clade Ia, Ib and clade IIb.

Corresponding Author: Dr Vincent Munster

Version 1:

Reviewer comments:

Reviewer #1

(Remarks to the Author)

A. Summary of the key results

The study reports the first confirmed case of MPXV clade IIb lineage A.2.2 in the Republic of the Congo using whole genome sequencing.

B. Originality and significance:

The study reports the first confirmed case of MPXV clade IIb lineage A.2.2 in the Republic of the Congo, which is a significant epidemiological finding considering the recent report of increased mpox cases in Western Africa caused by A.2.2 lineage. It also documents co-circulation of the two clades (I and II) and three subclades (Ia, Ib, IIb), which adds to our historical and current understanding of mpox virus evolution and spread in Central Africa. These findings are highly relevant for public health surveillance and preparedness across Africa irrespective of the clades that are currently reported.

C. Data & methodology: validity of approach, quality of data, quality of presentation

The researchers used well-established methodology for clinical diagnostics, quantitative real-time PCR (qRT-PCR), and whole-genome sequencing of MPXV. The dataset includes clinical, molecular, and genomic data from a confirmed mpox case, as well as passive surveillance data from 56 additional cases across seven administrative districts in the Republic of the Congo. The methodology is presented clear and detailed, outlining procedures for sample collection, sequencing protocols, and bioinformatics analysis. However, the 'Methods' section would benefit from proofreading to improve the overall flow and coherence of the narrative.

Additionally, lines 87–88 state that although the patient was isolated at home, there was “no transmission of the virus to family members or healthcare workers” but the authors did not provide enough detail to support this conclusion. It’s unclear whether this was based on regular follow-up testing, symptom monitoring, or active contact tracing. The frequency and scope of follow-up are not described, which makes it difficult to understand how secondary transmission was ruled out, especially in a region where in-house transmission is the highest. This is also important given that line 189 states the patient was unable to provide comprehensive contact-tracing information to health workers. These two statements seem contradictory and need to be clarified.

Although the manuscript is led by the National Public Health Laboratory and other relevant departments from the Ministry of Health and Population, the method section does not describe how the data generated were utilized or reported within the national mpox surveillance and response programs. Clarifying the link between the first detection of Clade II in the country and the public health response mechanisms would strengthen the manuscript and demonstrate the added value of genomics for outbreak management and response.

D. Appropriate use of statistics and treatment of uncertainties

The study is primarily descriptive and focused on bioinformatics and phylogenetic analysis; the authors used appropriate

computational tools for bioinformatics and phylogenetic. Cycle threshold (Ct) values are reported for qpCR results.

E. Conclusions: robustness, validity, reliability

The conclusions presented in the manuscript is well-supported by both genomic and epidemiological data. The phylogenetic placement of the mpox virus in lineage A.2.2 is robust (Figure 2), backed by appropriate analytical tools and sequence comparisons. Additionally, the use of well-validated diagnostic kits and standardized sequencing and bioinformatics protocols makes the result reliable. However, I encourage the authors to release the Whole genome sequences of the MPXV generated in this study. Line 226 does not provide the accession information.

F. Suggested improvements: experiments, data for possible revision

I believe the authors provided sufficient and currently available information to support the conclusions. The phylogenetic placement of the virus in lineage A.2.2 is clearly substantiated. However, the sequence data should be made public (if not yet).

G. References: appropriate credit to previous work?

Overall, the manuscript provides comprehensive citations to recent and relevant literature. However, Clade IIb has also been recently reported from DRC (WHO report:https://x.com/WHOAFRO/status/1953841921836683570?utm_source=chatgpt.com). It will be good to refer to this finding.

H. Clarity and context: lucidity of abstract/summary, appropriateness of abstract, introduction and conclusions

The abstract is clear and concise, summarizing the key findings and their implications. The introduction provides context on mpox epidemiology and clade distribution in the region, linking the study within the broader regional and global landscape. The discussion and conclusion sections carefully interpret the findings and highlight relevant public health implications.

Overall comment

The detection of MPXV clade IIb in the Republic of the Congo is a significant public health finding. However, the manuscript would benefit from further elaboration on how this detection was operationalized within the national mpox response (as mpox is a declared outbreak of continental and international concern). Specifically, it would be great if the authors clarify whether the case triggered any changes in surveillance protocols, reporting mechanisms, laboratory readiness, or public health interventions by the Ministry of Health or national emergency response teams. This will further strengthen the manuscript's relevance to public health policy and demonstrate how genomic data inform real-time public health decision-making.

Reviewer #2

(Remarks to the Author)

Although the manuscript submitted to Nature Medicine is timely and presents new data, its current scope remains limited and does not substantially advance the field of Mpox research. I would encourage the authors to strengthen the work by incorporating a more in-depth phylogenetic analysis. For example, molecular clock approaches could be used to infer the TMRCA of the introduction, which would help determine how recent the introduction is and whether the virus has been circulating undetected for some time. I also recommend performing phylogeographic analyses to better elucidate the potential origin of the introduction. Furthermore, it would be valuable to examine whether there are specific genomic changes compared to previously reported isolates and to assess whether any genes show evidence of selection. Incorporating these analyses would significantly enhance both the scientific contribution and the impact of the manuscript. Specific comments:

Line 131: "IG-TREE2" probably was supposed to be IQ-TREE

If you use ultrafast bootstrapping, there is no need to also perform SH-aLRT bootstrapping. Please choose one method (I would suggest ultrafast bootstrapping) and represent only that on the tree. In addition, kindly redraw the tree with branches arranged in increasing order (this option is available in FigTree).

The vertical branch at the root of the tree appears unusual—could the authors please re-root the tree. Please check and re-root or re-visualize appropriately.

Reviewer #3

(Remarks to the Author)

This manuscript presents the first laboratory-confirmed identification of mpox virus (MPXV) clade IIb lineage A.2.2 in the Republic of the Congo (RoC). This is an important and timely contribution that underscores both the rapidly evolving epidemiology of mpox in Central Africa and the continued global threat the virus poses. The methods are clearly described, and the discussion is generally well written and appropriately contextualized though there are suggestions that will strengthen the overall work.

Comments

- With multiple clades co-circulating in the same country, there is a tangible risk of genetic recombination between MPXV clades. Orthopoxviruses, including mpox, are well known for their recombination capacity. This potential evolutionary pathway could significantly alter the epidemiology of the disease and even lead to the emergence of novel recombinant

strains. The discussion should explicitly address this possibility and its public health implications.

- The authors do not clearly frame their results and it currently oscillates between being a case report and a broader surveillance report and gets confusing for the reader. The authors should clarify whether the central focus is the single case detection or the co-circulation of multiple clades. Both are important, but the manuscript would benefit from a clearer framing. The discussion ends with very generic recommendations for strengthening surveillance and vaccination. A stronger emphasis on region-specific barriers (e.g., diagnostic capacity, resource gaps, cross-border surveillance with DRC) would make the discussion more impactful. It is currently very boiler plate.

- Minor - typo line 178 German should be Germany, line 184 his should be this

Version 2:

Reviewer comments:

Reviewer #1

(Remarks to the Author)

The authors have thoroughly addressed all of my previous comments and concerns. The revisions made have significantly strengthened the manuscript in terms of clarity, structure, and overall scientific rigor. I have carefully reviewed the updated version and find that the improvements are substantial and satisfactory. At this stage, I have no additional comments or questions.

Reviewer #2

(Remarks to the Author)

Authors have satisfactorily addressed all comments, I don't have follow up comments.

Reviewer #3

(Remarks to the Author)

The authors have fully addresses the queries I raised in my initial review and revised the manuscript to my satisfaction . I have no additional edits or suggestions

We would like to thank the reviewers for their time and constructive comments. We feel that based on these comments our manuscripts improved markedly.

Referee expertise:

Referee #1: genomics, MPOX, molecular epidemiology, global health

Referee #2: genomics, phylogenetic analyses

Referee #3: ID clinician, policy

Reviewers' Comments:

Reviewer #1 (Remarks to the Author):

A. Summary of the key results

The study reports the first confirmed case of MPXV clade IIb lineage A.2.2 in the Republic of the Congo using whole genome sequencing.

B. Originality and significance:

The study reports the first confirmed case of MPXV clade IIb lineage A.2.2 in the Republic of the Congo, which is a significant epidemiological finding considering the recent report of increased mpox cases in Western Africa caused by A.2.2 lineage. It also documents co-circulation of the two clades (I and II) and three subclades (Ia, Ib, IIb), which adds to our historical and current understanding of mpox virus evolution and spread in Central Africa. These findings are highly relevant for public health surveillance and preparedness across Africa irrespective of the clades that are currently reported.

C. Data & methodology: validity of approach, quality of data, quality of presentation

The researchers used well-established methodology for clinical diagnostics, quantitative real-time PCR (qRT-PCR), and whole-genome sequencing of MPXV. The dataset includes clinical, molecular, and genomic data from a confirmed mpox case, as well as passive surveillance data from 56 additional cases across seven administrative districts in the Republic of the Congo. The methodology is presented clear and detailed,

outlining procedures for sample collection, sequencing protocols, and bioinformatics analysis. However, the 'Methods' section would benefit from proofreading to improve the overall flow and coherence of the narrative.

Response: Upon the reviewers suggestion we have proofread and modified this section accordingly to improve overall readability and flow

Additionally, lines 87–88 state that although the patient was isolated at home, there was “no transmission of the virus to family members or healthcare workers” but the authors did not provide enough detail to support this conclusion. It’s unclear whether this was based on regular follow-up testing, symptom monitoring, or active contact tracing. The frequency and scope of follow-up are not described, which makes it difficult to understand how secondary transmission was ruled out, especially in a region where in-house transmission is the highest. This is also important given that line 189 states the patient was unable to provide comprehensive contact-tracing information to health workers. These two statements seem contradictory and need to be clarified.

Response: according to the reviewer’s suggestion we have now modified the section to include more details on the isolation situation and the prevention of household transmission. The reviewer is correct that the patient was able to provide comprehensive contact-tracing information from the timepoint of likely exposure and infection but was able to provide of with information of the direct family members who were in contact with him. The section now reads:

“Home isolation set-up primarily comprised of voluntary isolation in separate housing from family members, with patient follow up every 3 days until lesion resolution. Health care workers who examined the patient and direct contacts of the patient were informed to contact the mpox disease control team in case of any observation of symptoms including myalgia, fever and skin lesions. No mpox symptoms were reported for any of the health workers or direct contacts.”

Although the manuscript is led by the National Public Health Laboratory and other relevant departments from the Ministry of Health and Population, the method section does not describe how the data generated were utilized or reported within the national mpox surveillance and response programs. Clarifying the link between the first detection of Clade II in the country and the public health response mechanisms would strengthen the manuscript and demonstrate the added value of genomics for outbreak management and response.

Response: upon the reviewers request we have now added additional information linking the detection with the public health response (which was already activated because of recent Clade 1b circulation in Point-Noire, RoC.

The section added to the discussion:

“The data generated on the detection of A.2.2. MPXV Clade IIb was directly shared with the RoC Directorate of Epidemiology and Disease Control of the Ministry of Health and Population under the framework of the Centre des Opérations des Urgences de Santé Publique (COUSP). This resulted in the implementation of Mpox control measures in Pointe-Noire including strengthening of local laboratory diagnostics using GeneXpert platforms to increase decentralized MPXV diagnostic testing, enhanced molecular epidemiology surveillance by NGS, outreach to hospitals, healthcare settings and healthcare workers to increase awareness on Mpox and to improve Mpox clinical management and hospital and healthcare hygiene practices with a focus on nosocomial transmission prevention.”

D. Appropriate use of statistics and treatment of uncertainties

The study is primarily descriptive and focused on bioinformatics and phylogenetic analysis; the authors used appropriate computational tools for bioinformatics and phylogenetic. Cycle threshold (Ct) values are reported for qPCR results.

E. Conclusions: robustness, validity, reliability

The conclusions presented in the manuscript is well-supported by both genomic and epidemiological data. The phylogenetic placement of the mpox virus in lineage A.2.2 is robust (Figure 2), backed by appropriate analytical tools and sequence comparisons. Additionally, the use of well-validated diagnostic kits and standardized sequencing and bioinformatics protocols makes the result reliable. However, I encourage the authors to release the Whole genome sequences of the MPXV generated in this study. Line 226 does not provide the accession information.

Response: we have now updated the manuscript to include the whole genome sequence data generated in this study.

F. Suggested improvements: experiments, data for possible revision

I believe the authors provided sufficient and currently available information to support the conclusions. The phylogenetic placement of the virus in lineage A.2.2 is clearly substantiated. However, the sequence data should be made public (if not yet).

Response: See above

G. References: appropriate credit to previous work?

Overall, the manuscript provides comprehensive citations to recent and relevant literature. However, Clade IIb has also been recently reported from DRC (WHO report: https://x.com/WHOAFRO/status/1953841921836683570?utm_source=chatgpt.com). It will be good to refer to this finding.

Response: We agree with the reviewer, however we could not find a source for proper citation. We have looked at the WHO website, the African CDC website and sources of the MoH of DRC, but none of these sites proved any source of confirmation of the social media posts.

H. Clarity and context: lucidity of abstract/summary, appropriateness of abstract, introduction and conclusions

The abstract is clear and concise, summarizing the key findings and their implications. The introduction provides context on mpox epidemiology and clade distribution in the region, linking the study within the broader regional and global landscape. The discussion and conclusion sections carefully interpret the findings and highlight relevant public health implications.

Overall comment

The detection of MPXV clade IIb in the Republic of the Congo is a significant public health finding. However, the manuscript would benefit from further elaboration on how this detection was operationalized within the national mpox response (as mpox is a declared outbreak of continental and international concern). Specifically, it would be great if the authors clarify whether the case triggered any changes in surveillance protocols, reporting mechanisms, laboratory readiness, or public health interventions by the Ministry of Health or national emergency response teams. This will further strengthen the manuscript's relevance to public health policy and demonstrate how genomic data inform real-time public health decision-making.

Response, we agree with the reviewer and have updated our manuscript accordingly (see above).

Reviewer #2 (Remarks to the Author):

Although the manuscript submitted to Nature Medicine is timely and presents new data, its current scope remains limited and does not substantially advance the field of Mpox research. I would encourage the authors to strengthen the work by incorporating a more in-depth phylogenetic analysis. For example, molecular clock approaches could be used to infer the TMRCA of the introduction, which would help determine how recent the introduction is and whether the virus has been circulating undetected for some time. I also recommend performing phylogeographic analyses to better elucidate the potential origin of the introduction. Furthermore, it would be valuable to examine whether there are specific genomic changes compared to previously reported isolates and to assess whether any genes show evidence of selection. Incorporating these analyses would significantly enhance both the scientific contribution and the impact of the manuscript.

Response: We thank the reviewer for these thoughtful and constructive comments, which have helped us to substantially strengthen the manuscript. In response, we have conducted an expanded phylogenetic and molecular clock analysis using BEAST to estimate the substitution rate and infer the timing of the viral introduction into the Republic of the Congo (RoC). The estimated mean rate of evolution of the clade was 5.5×10^{-5} substitutions/site/year, consistent with values previously reported for Mpox clade IIb lineage A.2. The time to the most recent common ancestor (TMRCA) between the RoC genome (hMpxV/Congo/BZV-LNSP-CG051/2025) and its closest non-RoC relatives (from the United States and Australia) was estimated at mid-2023 (mean 2023.5; 95% HPD 2021.44–2024.63). Based on this, we infer that the lineage leading to the RoC strain originated between August 2024 and March 2025 .

Finally, we acknowledge that only a single genome is currently available from the Republic of the Congo (RoC), which limits our ability to perform robust phylogeographic inference or to determine the precise geographic origin of this introduction. Moreover, there are very few genomes globally in this cluster of the lineage A.2.2, and only a handful from Africa overall, further constraining efforts to trace the evolutionary and geographic pathways of this lineage. This paucity of data makes it difficult to pinpoint the spillover origin or to assess whether undetected circulation has occurred regionally. We have revised the Discussion to explicitly highlight these limitations and to emphasize the urgent need for enhanced genomic surveillance in Central Africa and other underrepresented regions to enable more accurate phylogeographic and evolutionary inference in future studies.

Specific comments:

Line 131: “IG-TREE2” probably was supposed to be IQ-TREE

If you use ultrafast bootstrapping, there is no need to also perform SH-aLRT bootstrapping. Please choose one method (I would suggest ultrafast bootstrapping) and

represent only that on the tree. In addition, kindly redraw the tree with branches arranged in increasing order (this option is available in FigTree).

Response: We have corrected IG-TREE2 to IQ-TREE. In addition, we have used ultrafast bootstrapping, as suggested by the reviewer.

The vertical branch at the root of the tree appears unusual—could the authors please re-root the tree. Please check and re-root or re-visualize appropriately.

Response: This has been corrected.

Reviewer #3 (Remarks to the Author):

This manuscript presents the first laboratory-confirmed identification of mpox virus (MPXV) clade IIb lineage A.2.2 in the Republic of the Congo (RoC). This is an important and timely contribution that underscores both the rapidly evolving epidemiology of mpox in Central Africa and the continued global threat the virus poses. The methods are clearly described, and the discussion is generally well written and appropriately contextualized though there are suggestions that will strengthen the overall work.

Comments

- With multiple clades co-circulating in the same country, there is a tangible risk of genetic recombination between MPXV clades. Orthopoxviruses, including mpox, are well known for their recombination capacity. This potential evolutionary pathway could significantly alter the epidemiology of the disease and even lead to the emergence of novel recombinant strains. The discussion should explicitly address this possibility and its public health implications.

Response: upon the reviewers suggestion, we have updated the discussion to include a paragraph of the potential for recombination of MPXV.

Mpox virus evolution is driven by a variety of mechanisms and underlies the relatively rapid genotypic and phenotypic change observed during the consecutive mpox outbreaks of the last decade. These mechanisms include acquiring mutations (e.g. the APOBEC3 mutations), gene duplication and recombination. The co-circulation of clade Ia, Ib and clade IIb in the human population of a relatively confined geographical area dramatically increases the risk for recombination events between viruses of these different clades, potentially resulting new phenotypes.

- The authors do not clearly frame their results and it currently oscillates between being a case report and a broader surveillance report and gets confusing for the reader. The

authors should clarify whether the central focus is the single case detection or the co-circulation of multiple clades. Both are important, but the manuscript would benefit from a clearer framing.

Response: we agree with the reviewer that the manuscript is a in a bit of a hybrid form, reflecting both the initial detection as well as putting the work into the larger context of the ongoing MPXV surveillance in the RoC. We have tried to improve the overall readability but have kept the hybrid form.

The discussion ends with very generic recommendations for strengthening surveillance and vaccination. A stronger emphasis on region-specific barriers (e.g., diagnostic capacity, resource gaps, cross-border surveillance with DRC) would make the discussion more impactful. It is currently very boiler plate.

Response: we agree with the reviewer and have now updated this paragraph accordingly:

The co-circulation of clade Ia, Ib and clade IIb in the human population highlights the increasing complexity of Mpox epidemiology. The detection of co-circulation of multiple clades in the Congo basin highlights the need of a regionally coordinated response. This response should be targeted at region specific needs, including expanding access to decentralized diagnostics, enhancing the (molecular) epidemiological capacity and dramatic increase in vaccine access. Implementation of isolation and contact-tracing strategies, as well as community outreach should be a key component of a targeted Mpox control strategy. Lastly, our ability to detect emerging and re-emerging pathogens is directly related to the accessibility to healthcare and diagnostic infrastructure. Large parts of the RoC are relatively remote and more emphasis on increased surveillance in border regions between RoC, DRC and CAR are needed.

- Minor - typo line 178 German should be Germany, line 184 his should be this

Response: This has been corrected.